# Neutralizing monoclonal antibodies improve biodistribution of intravenously administered oncolytic adenovirus in human CD46-transgenic mice

**Masahisa Hemmi**, **Midori Yamashita**, **Yoshiko Shimizu, Shinsuke Nakao**\*

Immuno-Oncology, Astellas Pharma Inc., Ibaraki, Japan

\* shinsuke.nakao@astellas.com

## Abstract

Oncolytic viruses are a unique modality with multifaceted mechanisms of action for killing cancer cells and have been developed as a promising therapeutic approach in cancer treatment. The first-in-class agent, talimogene laherparepvec (T-VEC), has shown clinical benefit in patients with advanced melanoma. However, intratumoral administration of oncolytic viruses has several limitations which prevent use against a broader range of cancer types. Here, we propose a novel treatment strategy consisting of the intravenous administration of a genetically engineered oncolytic adenovirus type 11 (Ad11) mixed with anti-Ad11 neutralizing monoclonal antibodies. Ad11, which has minimum binding affinity to human erythrocytes, was modified to selectively replicate in cancer cells. New anti-Ad11 antibody clones were generated which inhibit binding between Ad11 fibers and their natural receptor, CD46. The neutralizing antibodies suppressed viral accumulation in the lungs by about 10-fold in human CD46-transgenic mice without loss of infectivity to cancer cells. Our findings are important in ensuring safe and efficient virus delivery following intravenous administration in humans and may expand treatment options.

## Introduction

Although cancer treatment options have dramatically increased, effective treatments for advanced solid tumors remain to be developed, and new agents with unique mechanisms of action are currently under evaluation. Among these, oncolytic virotherapy is a promising immunotherapeutic modality for various types of tumors [1,2]. Following intratumoral administration, oncolytic viruses (OVs) selectively replicate in the tumor and induce antitumor immune responses to kill cancer cells [3–5]. Intratumoral administration of talimogene laherparepvec (T-VEC), a genetically modified herpes simplex virus type 1 encoding granulocyte macrophage colony-stimulating factor (GM-CSF), has proved its clinical benefit for patients with advanced

**Data availability statement:** All relevant data are within the manuscript and its Supporting Information files.

**Funding:** This research was funded by Astellas Pharma, Inc..

**Competing interests:** I have read the journal's policy and the authors of this manuscript have the following competing interests: the authors are employees of Astellas Pharma Inc., Japan.

melanoma [6–9]. Intratumoral administration ensures direct delivery of OVs to the targeted tumor. However, physicians cannot always inject OVs into every tumor, particularly when tumors are spread throughout the body or when they are located in inaccessible visceral areas [10]. These untreated tumors can be attacked by in situ vaccination evoked in virus-injected tumors [4,5]. Efficacy is not always sufficient, especially if the metastases are not immunogenic [11] In this context, intravenous administration is more practical for clinical use and holds greater potential in fighting disseminated tumors, thereby extending patient survival [12].

Intravenous administration raises concerns about adequate virus delivery to tumors. One reason is the rapid clearance of OVs by pre-existing neutralizing antibodies during circulation [13,14]. Viruses with lower seroprevalence are more suitable, such as vaccinia virus, which was used until the 1980s in a worldwide vaccination campaign to eradicate the smallpox virus [15]. A clinical study of an oncolytic vaccinia virus, pexastimogene devacirepvec (Pexa-Vec), which was administered intravenously to patients with colorectal cancer liver metastases or metastatic melanoma, confirmed the existence of the virus in analyzed tumors [16,17]. Another option is the use of rare adenovirus serotypes from groups B (Ads 3, 11, 35, etc.) or D (Ads 26, 28, 51, etc.) [18–20]. A study of enadenotucirev, a chimeric oncolytic adenovirus with the capsid from Ad11p, was confirmed to produce viral delivery to tumors after intravenous administration [21–24]. Nevertheless, the unwanted uptake of OVs by normal tissues rather than the targeted tumors remains an issue not only with regard to the efficiency of delivery but also safety. Vaccinia virus is permissive for most mammalian cell types, including endothelial cells of blood vessels [25–27]. Adenovirus serotypes including Ad5 rapidly distribute to the liver, mediated by coagulation factor X (FX), which links the Ad5 hexon to hepatocytes, leading to acute hepatotoxicity [28,29]. Although Ads 3, 11 and 35 have weak binding to FX and Ads 26, 28 and 51 show no binding, their natural tropism for CD46, which is commonly expressed on all nucleated cells in humans, reduces virus delivery to tumors [19,30,31].

With the aim of intravenous administration to patients with systemic metastases, we engineered a recombinant oncolytic Ad11 using a genetic design to selectively replicate in cancer cells with abnormal Rb activity [32,33]. Additionally, we generated anti-Ad11 monoclonal antibodies to interrupt binding to CD46. Intravenous administration of Ad11 mixed with monoclonal antibodies showed an approximately 10-fold decrease in virus accumulation in normal organs, including the lung, in human CD46-transgenic (CD46tg) mice. To our knowledge, this is the first description of a method to alter the tropism of Ad11 in mice utilizing a single monoclonal antibody clone. This approach could be extended to other viruses utilizing CD46 and may represent a novel therapeutic option in patients with systemic metastases.

## Materials and methods

### Cell lines, cell culture, and stable cell line

A549, HT-29, Panc-1, PC-3, HeLa, Hep3B, WI-38, BxPC-3, MDA-MB-231, SK-OV-3, SW480, MRC-5, WI-38 and CHO cells were purchased from the American Type Culture Collection (ATCC); HEK293 (Adeno-X 293) cells from Clontech; HUVEC from

Lonza; and MKN45 cells from RIKEN Cell Bank. Cells were maintained in humidified incubators at 37°C and 5% $CO_2$ in the indicated media supplemented with 10% fetal bovine serum (FBS; GE Healthcare) and 1% penicillin-streptomycin (PS; Thermo Fisher Scientific). HEK293, A549, Panc-1, PC-3, HeLa, Hep3B and WI-38 cells were cultured in Dulbecco's modified Eagle's medium (DMEM; Sigma-Aldrich); BxPC-3, MDA-MB-231, SK-OV-3, MKN45 and SW480 cells in Roswell Park Memorial Institute (RPMI)-1640 medium (Sigma-Aldrich); HT-29 cells in McCoy's 5A medium (Thermo Fisher Scientific); HUVEC in HuMedia-EG2 (KURABO); MRC-5 cells in Minimum Essential Medium (MEM; Sigma-Aldrich) additionally supplemented with 2 mM L-glutamine (Sigma-Aldrich) and 1 × NEAA (Thermo Fisher Scientific); CHO cells in F-12 Ham medium (Sigma-Aldrich); and human CD46-expressing CHO cells in F-12 Ham medium supplemented with 2 mg/mL G418 (Thermo Fisher Scientific). All cells were tested to be mycoplasma-free.

Human CD46-expressing CHO cells were generated by Astellas Pharma Inc. The pCMV6-CD46 Human (Accession No. NM_002389) Tagged ORF Clone (Origene) was transfected into CHO cells using Lipofectamine 2000 (Thermo Fisher Scientific) to express human CD46. The CD46-expressing cells were sorted, and stable transfectants were obtained and maintained under selection pressure using G418.

### Wild-type viruses

Wild-type adenoviruses were purchased from ATCC. Virus was propagated in HEK293 cells and then purified using standard techniques. Briefly, adenovirus was purified by $CsCl_2$ step gradient ultracentrifugation, dialyzed with phosphate-buffered saline (PBS; Nacalai Tesque) containing 10% UltraPure Glycerol (Thermo Fisher Scientific), and stored in aliquots at −80 °C. Virus particle (vp) titers were spectrophotometrically quantified by measuring the optical density at 260 nm ($OD_{260}$).

### Recombinant viruses

Recombinant Ad11 genomic DNA was constructed by a DNA assembly technology using NEBuilder HiFi DNA Assembly Master Mix (New England Biolabs). PCR primers were designed to amplify the partial Ad11 sequence and add overlap sequences at both the 5' and 3' ends for DNA assembly. Viral genomic DNA was extracted from wild-type Ad11 (GenBank sequence ID: AF532578.1) for use in the PCR template. In addition, the modified Ad11 E1A sequence which contained the E2F1 promoter region and the Ad11 E1A gene with Δ24 mutation, and the modified fiber sequence which contained mutation in the Ad11 fiber knob were artificially synthesized using a custom DNA oligo service (Thermo Fisher Scientific). To obtain the vector plasmid of Ad11-E2F-Δ24, PCR fragments of Ad11 were assembled in a step-by-step manner using NEBuilder HiFi DNA Assembly Master Mix. Next, expression cassettes containing the murine IL-12 gene, cytomegalovirus (CMV) promoter-driven enhanced green fluorescent protein (EGFP) gene, or CMV promoter-driven murine IL-12 gene were inserted into the E3-deleted region of Ad11-E2F-Δ24 by the DNA assembly technology. To generate recombinant viruses, the assembled DNA vector was transfected into HEK293 cells using Lipofectamine 2000, and then the viruses were propagated and purified as described above.

### Animals

All animal procedures and experiments were approved by the Institutional Animal Care and Use Committee of Astellas Pharma Inc., Tsukuba Research Center, which is accredited by AAALAC (Association for Assessment and Accreditation of Laboratory Animal Care) International. All animal experiments were conducted in accordance with institutional ethical guidelines. Mice were anesthetized using isoflurane to minimize pain and distress. Mice were humanely sacrificed by carbon dioxide inhalation in a gradually filled chamber to ensure a humane endpoint. Efforts were made to alleviate suffering by closely monitoring animals for signs of pain or distress, and appropriate interventions were applied when necessary. C57BL/6, Balb/c, hCD46-tg mice were purchased from Charles River Laboratories Japan Inc., and SCID mice from Jackson Laboratory Japan Inc. They were maintained on a standard diet and water ad libitum throughout the experiments under

specific pathogen–free conditions. The plasma aspartate aminotransferase (AST) and alanine aminotransferase (ALT) levels were determined using mouse AST and ALT ELISA kits (Abcam). Cynomolgus monkeys (Macaca fascicularis) were purchased from Hamri Co., Ltd. All monkeys were free of simian immunodeficiency virus, salmonella bacteria, dysentery bacteria, and B virus. They were housed in individual cages, allowed free access to water, and given food once a day.

## Hemagglutination assay

Human whole blood samples were obtained from healthy adults aged over 20 years who understood the purpose of this research and participated voluntarily. No information was available on antibody titers for each serotype. The study was approved by Astellas Research Ethics Committee. The recruitment period for the study is 17th Apr 2019–31st Mar 2025. The participants provided informed consent in a written form. Freshly collected human or monkey whole blood was used. Blood samples were layered carefully onto Ficoll-Paque (GE Healthcare) in a 15 mL tube. The samples were centrifuged at 400 × g for 30 minutes at 20 °C. After centrifugation, the plasma, platelet, white blood cell, and granular cell layers were carefully removed. The red blood cells (RBCs) were then collected, washed with 5 mM EDTA (Invitrogen)/PBS, and centrifuged again. The wash step was repeated once. The washed red blood cells were resuspended in 5 mM EDTA/PBS to achieve a final concentration of 1% (v/v). Fifty microliters of the 1% (v/v) red blood cell suspension were added to each well of a round-bottom 96-well plate. Subsequently, 50 µL of the serially diluted virus samples were added to each well and mixed gently. Wild-type adenoviruses were used for the human assay. Wild-type Ad11, Ad11-muFiber, and Ad11-E2F-Δ24-CMV-mIL12 were used for the monkey assay. The plate was incubated at 37 °C for 2–3 hours. After incubation, the hemagglutination patterns were observed. In the absence of adenovirus interaction with RBCs, hemagglutination does not occur and RBCs form a small and rounded dot. In the presence of adenovirus interaction with RBCs, RBCs and adenoviruses form complexes, resulting in hemagglutination that appears as a large, blurred sediment.

## In Vitro Cytotoxicity Assay

Cells were seeded at a density of $1 \times 10^4$ cells per well in a 96-well plate. The following day, virus solution was added at a concentration of $1 \times 10^4$ vp/cell in a 10-fold serial dilution. The cells were then incubated for 4 days. After the incubation, CellTiter-Glo 2.0 reagent (Promega) was added to each well. The plate was vortexed briefly and allowed to sit at room temperature for 10 minutes to stabilize the luminescent signal. The mixture from each well was then transferred to a 96-well white plate, and luminescence was measured using an ARVO X3 luminometer (PerkinElmer). Cell viability was calculated by setting uninfected cells (negative control) and medium control wells (positive control) to 100% and 0% survival, respectively.

## In Vitro replication efficiency assay

Cells were seeded at a density of $1 \times 10^4$ cells per well in a 96-well plate. The following day, cells were infected with Ad11, Ad28 and Ad51 at a concentration of 100 and 1000 vp/cell for 2 hours. After infection, the cells were washed with PBS once and fresh culture medium was added. The cells were collected at 24 and 72 hours after infection. The samples were stored at −80 °C until DNA extraction for qPCR analysis.

## In Vitro IL-12 detection assay

Cells were seeded at a density of $1 \times 10^4$ cells per well in a 96-well plate. The following day, cells were infected with Ad11-E2F-Δ24-mIL12 at a concentration of 100 vp/cell for 2 hours. After infection, the cells were washed with PBS once and fresh culture medium was added. The supernatant was collected at 24 hours after infection. Mouse IL-12 level in the supernatant was detected with a mouse IL-12 p70 Duoset ELISA kit (R&D Systems), and absorbance was measured using an ARVO X3 luminometer (PerkinElmer).

## In vitro virus binding assay and in vivo biodistribution study

For in vitro virus binding assay, CHO or human CD46-expressing CHO cells were seeded at a density of $6 \times 10^4$ cells per well in a 12-well plate. The following day, the supernatant was discarded, and the cells were washed with PBS. Wild-type adenoviruses was added at a concentration of 15 vp/cell and the plate was incubated at 4 °C for 2 hours. After incubation, the cells were washed with PBS three times and then collected for qPCR analysis to detect viral DNA. For in vivo biodistribution study, mice were intravenously injected with adenoviruses, and then lung, liver, spleen and blood were collected. The samples were suspended in PBS using the gentleMACS Dissociator (Miltenyi Biotec) and the lysates were subjected to qPCR analysis.

For qPCR analysis, total DNA was isolated from cells or organ lysates using a QIAamp DNA mini kit (QIAGEN). Quantitative PCR was performed with TaqMan Gene Expression Master Mix and a TaqMan MGB gene expression detection set (Applied Biosystems) using Quant Studio12K Flex (Thermo Fisher Scientific). Absolute quantities were calculated using standard curves. For biodistribution experiments, the copy number of each gene was normalized with total DNA amount in the organ. The primer and probe sequences are described in S1 Table.

## Generation of Anti-Ad11 monoclonal antibodies

Transgenic rodents (AlivaMab mice; Ablexis) were intramuscularly immunized with wild-type Ad11 repeatedly. About 1–2 months later, lymph nodes were collected. Lymphocytes isolated from lymph nodes were fused with a mouse myeloma cell line Sp2/0 (ATCC, #CRL-1581) using GenomeONE (CosmoBio) to make hybridomas. Supernatants from the hybridomas were screened according to their affinity to Ad11 and the inhibitory activity to Ad11 infection as described below.

Total RNA of each candidate clone was extracted from the hybridoma using RNeasy kit (QIAGEN). First-strand cDNA was synthesized using a SMARTer RACE cDNA Amplification kit (Takara Bio). The anti-Ad11 antibody DNA fragment was amplified by adding a restriction enzyme site and inserted into the expression vector pcDNA3.4-TOPO (Thermo Fisher Scientific) with a constant region of mutated human immunoglobulin g1 chain (L234A/L235A/P331G) using HindIII and NheI. The light chain variable region was inserted into the expression vector pcDNA3.4-TOPO (Thermo Fisher Scientific) with the constant region of human kappa or lambda chain using HindIII and BsiWI or EcoRI. A DNA construct for anti-Ad11 antibody was expressed in ExpiCHO™ cells (Thermo Fisher Scientific) and secreted antibody was purified by Protein A affinity chromatography (Cytiva). Eluted material was buffer exchanged into PBS pH 7.4 (Thermo Fisher Scientific).

## GFP assay

HEK293 cells were seeded at a density of $5 \times 10^3$ cells per well in a 384-well plate. The following day, Ad11-E2F-Δ24-CMV-EGFP was incubated with each culture supernatant of hybridomas, mouse antiserum, or purified anti-Ad11 monoclonal antibody 4G1 at 37 °C for 1 hour. The incubated virus solutions were infected to the cells at a concentration of 30 vp/cell for 24 hours. GFP signal was detected using the IncuCyte S3 live cell imaging system (Sartorius). GFP count was calculated using IncuCyte software (Sartorius).

## ELISA for anti-Ad11 antibody

96-well microtiter plates were coated with $2.5 \times 10^7$ vp of Ad11-E2F-Δ24-CMV-mIL12 and incubated overnight at 4°C. The wells were blocked with Blocking One (Nacalai Tesque) at room temperature for 1 hour. They were treated with diluted anti-Ad11 antibodies and incubated at room temperature for 1 hour and then treated with horseradish peroxidase-conjugated anti-human IgG antibody (diluted 1:5000, Jackson Laboratories). Bound IgG was detected with a TMB substrate reagent set (BD Biosciences), and absorbance was measured using an ARVO X3 luminometer (PerkinElmer).

## Whole blood binding assay

To measure the amount of virus bound to blood cells, 50 µL of whole blood was added to each well of a U-bottom 96-well plate. Five microliters of $1 \times 10^7$ vp of wild-type Ad11, Ad11-mufiber, and Ad11-E2F-$\Delta$24-CMV-mIL12 were then added to each well, following incubation with antibodies for 1 hour at 37 °C. Whole blood cells were infected and incubated at 37 °C for 30 minutes. Then, the samples were washed with PBS and centrifuged at $400 \times g$ at 4 °C for 5 minutes. The supernatant was carefully removed. This washing step was repeated twice. The samples were stored at −80 °C until DNA extraction.

## Western blotting analysis

Lung, liver and spleen collected from mice were homogenized in T-PER buffer (Thermo Fisher Scientific) and cOmplete (Sigma), and the supernatants were then obtained after centrifugation of the homogenates. The concentrations of total protein in these samples were determined using the BCA Protein Assay Kit (Thermo Fisher Scientific). The samples (7.5 µg protein per sample) were lysed with LDS sample buffer (Thermo Fisher Scientific) and then resolved by SDS-PAGE and transferred to PVDF membranes. After blocking at room temperature with Blocking One (Nacalai Tesque), the membranes were incubated overnight with the following primary antibodies: human CD46 (clone 4C3, Bioss Antibodies) and mouse b-actin (clone C-2, Santa Cruz). After washing with TBS-T, membranes were incubated with horseradish peroxidase-conjugated secondary antibodies (Cell Signaling Technology) for 1 hour at room temperature. Proteins of interest were visualized by enhanced chemiluminescence using the ECL Prime Western Blotting Detection System (GE Healthcare) and detected with LAS-4000 (GE Healthcare).

## Anti-tumor efficacy study

For the anti-tumor efficacy study, immunocompromised SCID mice bearing the human cancer cell line MKN45 were used. Cells ($2 \times 10^6$ per mouse) were subcutaneously inoculated into the right flank of the mice. When tumors reached about 50–100 mm³, intravenous treatment with Ad11-E2F-$\Delta$24-CMV-mIL12 or the mixture of Ad11-E2F-$\Delta$24-CMV-mIL12 and 4G1 was initiated and repeated every other day for a total of three doses. Diameters of subcutaneous tumors were measured using digital calipers, and tumor volume was calculated by the formula length $\times$ width$^2$ $\times$ 0.5. Tumor sizes and body weights of individual animals were continuously monitored. During the study, mice were monitored daily by staffs with specialized training in animal care and handling. The care and use of mice was in accordance with the principles outlined in the Guide for the Care and Use of Laboratory Animals 8th Edition (National Research Council). When signs of deterioration or acute weight loss were observed, or when the average tumor diameter exceeded 1.5 cm, mice were euthanized on the same day. No mice died before meeting criteria in the study.

## Statistical analysis

Statistical analysis was conducted using GraphPad Prism 10 (GraphPad Software). Procedures used for comparison and the numbers of animals in the experiment are described in each figure. $P$ values <0.05 were considered to be significant. All results are shown as the mean ± SEM (biological replicates) or SD (technical replicates).

# Results

### Ad11 kills human cancer cells and has no affinity for human RBCs

To select a suitable adenovirus serotype to be engineered for intravenous administration, various serotypes from different groups were characterized. First, Ad11 (Group B), Ad5 (Group C), Ad26, Ad28 and Ad51 (Group D) were subjected to hemagglutination assay [34] to exclude serotypes that bind to human RBCs, considering that the binding to RBCs potentially reduces the delivery efficiency after intravenous administration (Fig 1A). Experiments revealed that Ad5 interacted with human RBCs from all donors (n = 5), while Ad26 interacted with RBCs from two donors (Fig 1B). Ad11, Ad28 and

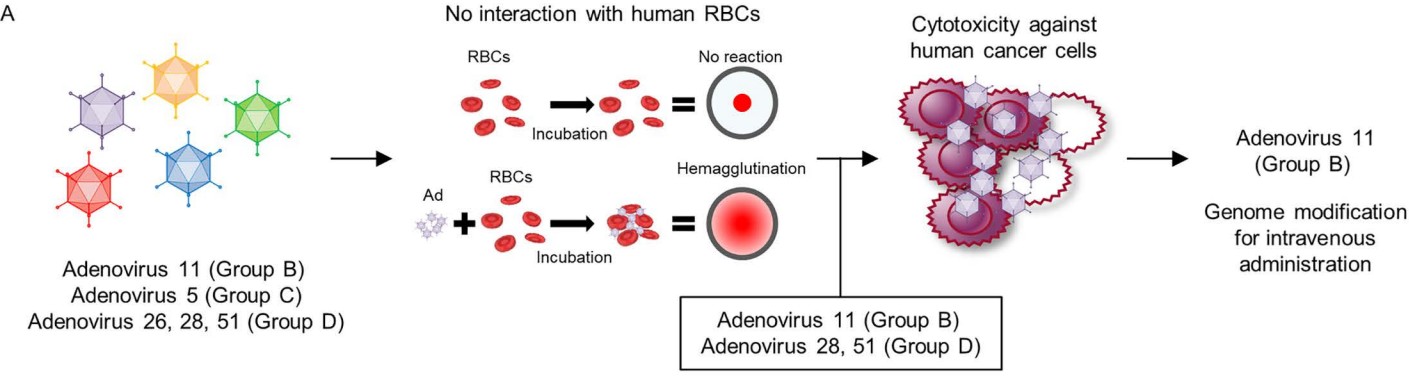

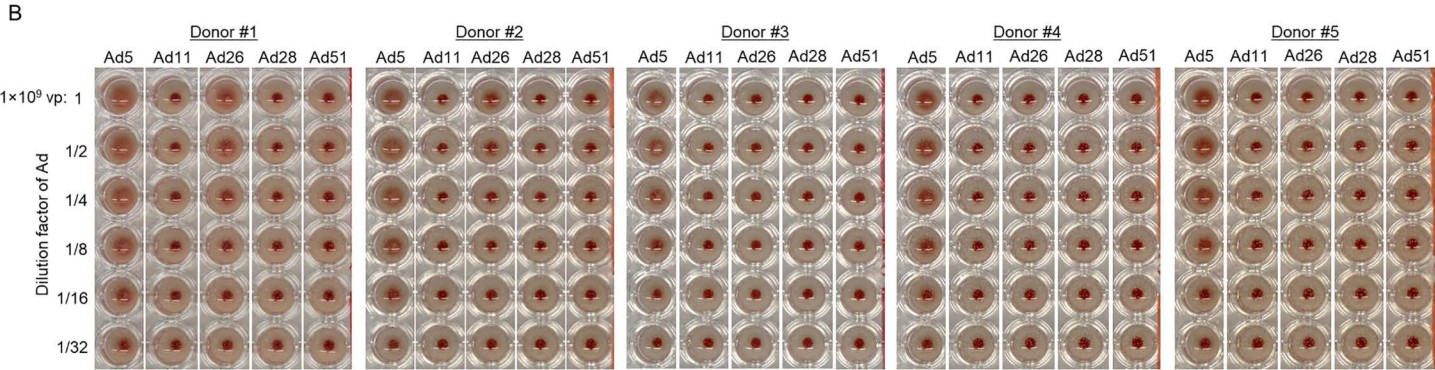

**Fig 1. Selection of human adenovirus serotypes with no affinity to human RBCs. (A)** Schematic representation of the selection from wild-type adenovirus 11 (Group **B**), adenovirus 5 (Group **C**), and adenovirus 26, 28, and 51 (Group D) in Figs 1 and 2. First, adenovirus interaction with human RBCs was assessed by hemagglutination assay. Second, adenovirus serotypes that did not interact with human RBCs were evaluated for their cytotoxicity against human cancer cell lines. Ad11, which exhibited broad cytotoxicity against human cancer cell lines, was selected for genome modification for intravenous administration. **(B)** Human RBCs (5 donors) were incubated with each wild-type adenovirus. After sedimentation of RBCs finished, hemagglutination patterns were observed.

Ad51 showed no interaction. Next, cytotoxicity of Ad11, Ad28 and Ad51 against human cancer cell lines was assessed. The ratio of vp/TCID$_{50}$ for Ad11, Ad28 and Ad51 used in this experiment was 1:1.3:0.3. Hepatocellular carcinoma Hep3B, colorectal adenocarcinoma HT29, and prostatic adenocarcinoma PC-3 showed sensitivity to Ad11, Ad28 and Ad51 with different potency, with Ad11 showing the strongest cytotoxicity (Fig 2). Likewise, Ad11 showed cytotoxicity against lung carcinoma A549, ovarian adenocarcinoma SK-OV-3, breast adenocarcinoma MDA-MB-231, and pancreatic carcinoma Panc-1, whereas the cytotoxicity of Ad28 and Ad51 against these cell lines was moderate or minimum. At 72 hours post-infection, Ad11 showed higher viral copy numbers in A549 cells compared to Ad28 and Ad51, and comparable or lower in Hep3B, PC-3 and HT-29 cells (S1A Fig). From 24 to 72 hours, Ad11 demonstrated a greater increase in viral copy numbers across all cell lines, suggesting a more efficient replication capacity than Ad28 and Ad51 (S1B Fig). This enhanced replication may contribute to the superior cytotoxicity observed with Ad11. BxPC-3 showed less sensitivity to Ad11, Ad28 and Ad51. Considering these results together, we selected Ad11 for further engineering.

## Ad11 adoption of the E2F1 promoter and E1A modification demonstrates selectivity for cancer cells

In several oncolytic adenoviruses currently under clinical investigation, the E1A promoter has been replaced with the human E2F1 promoter, allowing virus replication to be restricted only in cells that are defective in the Rb-pathway. This

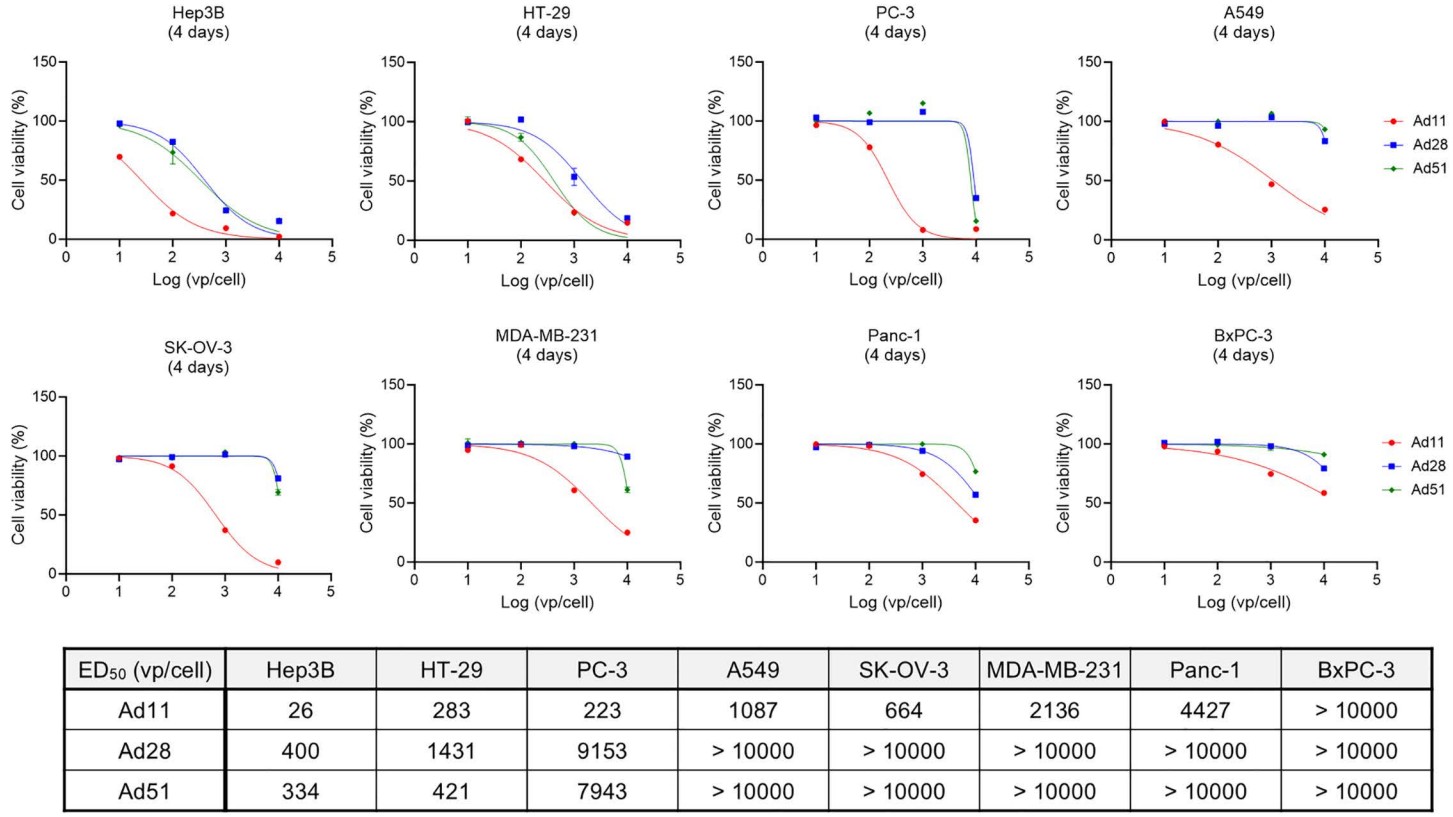

| ED$_{50}$ (vp/cell) | Hep3B | HT-29 | PC-3 | A549 | SK-OV-3 | MDA-MB-231 | Panc-1 | BxPC-3 |
|---|---|---|---|---|---|---|---|---|
| Ad11 | 26 | 283 | 223 | 1087 | 664 | 2136 | 4427 | > 10000 |
| Ad28 | 400 | 1431 | 9153 | > 10000 | > 10000 | > 10000 | > 10000 | > 10000 |
| Ad51 | 334 | 421 | 7943 | > 10000 | > 10000 | > 10000 | > 10000 | > 10000 |

**Fig 2. Screening of human adenovirus serotypes to select a serotype with broad cytotoxicity against human cancer cell lines.** Different types of human cancer cell lines were infected with wild-type adenovirus serotypes at different titers (vp/cell). Four days after viral infection, cytotoxicity against each cell line was measured. The experiment was conducted in triplicate. Mean±SD is shown. Table indicates the half-maximal effective dose (ED$_{50}$) values for each cell line.

modification is based on the fact that E1A encodes essential proteins required early in the virus replication cycle [32,35]. Additionally, some of these have a 24 base-pair deletion (Δ24) in the E1A gene encoding the Rb-binding domain to avoid self-activation of E2F1 promoter mediated by E1A-Rb interaction [33,36,37]. Likewise, Ad11 was engineered to have the E2F1 promoter and Δ24 (Ad11-E2F-Δ24) (Figs 3A-3C). Ad11-E2F-Δ24 did not lose its viral fitness in cytotoxicity against cancer cells (S2 Fig). Ad11-E2F-Δ24 at $10^3$ vp/cell demonstrated 100% cytotoxicity against human gastric adenocarcinoma MKN45 and human colorectal adenocarcinoma SW480 cancer cell lines, while exhibiting minimal effects on normal human fibroblast MRC-5 (Fig 3D). Ad11-E2F-Δ24 at $10^4$ vp/cell reached less than 50% cytotoxicity against MRC-5. MKN45 infected with 10 vp/cell of Ad11-E2F-Δ24-IL12, which has murine IL-12 gene insertion into the E3 region of Ad11-E2F-Δ24 (Fig 3C), secreted a greater than 30-fold higher level of IL-12 protein compared to normal cells, including MRC-5, HUVEC (normal human umbilical vein endothelial cells) and WI-38 (normal human fibroblast) (Fig 3E). These results indicate that Δ24 and E2F1 promoter designed in the recombinant Ad11 successfully control viral replication in normal cells.

## Neutralizing monoclonal antibodies alter biodistribution of intravenously administered Ad11 in human CD46 transgenic mice

Ad11 and some Ads from Group D use human CD46 as a primary receptor for infection, whereas Ad5 uses Coxsackievirus-Adenovirus Receptor (CAR) (Fig 4A). Because CD46 expression occurs in all nucleated cells in humans,

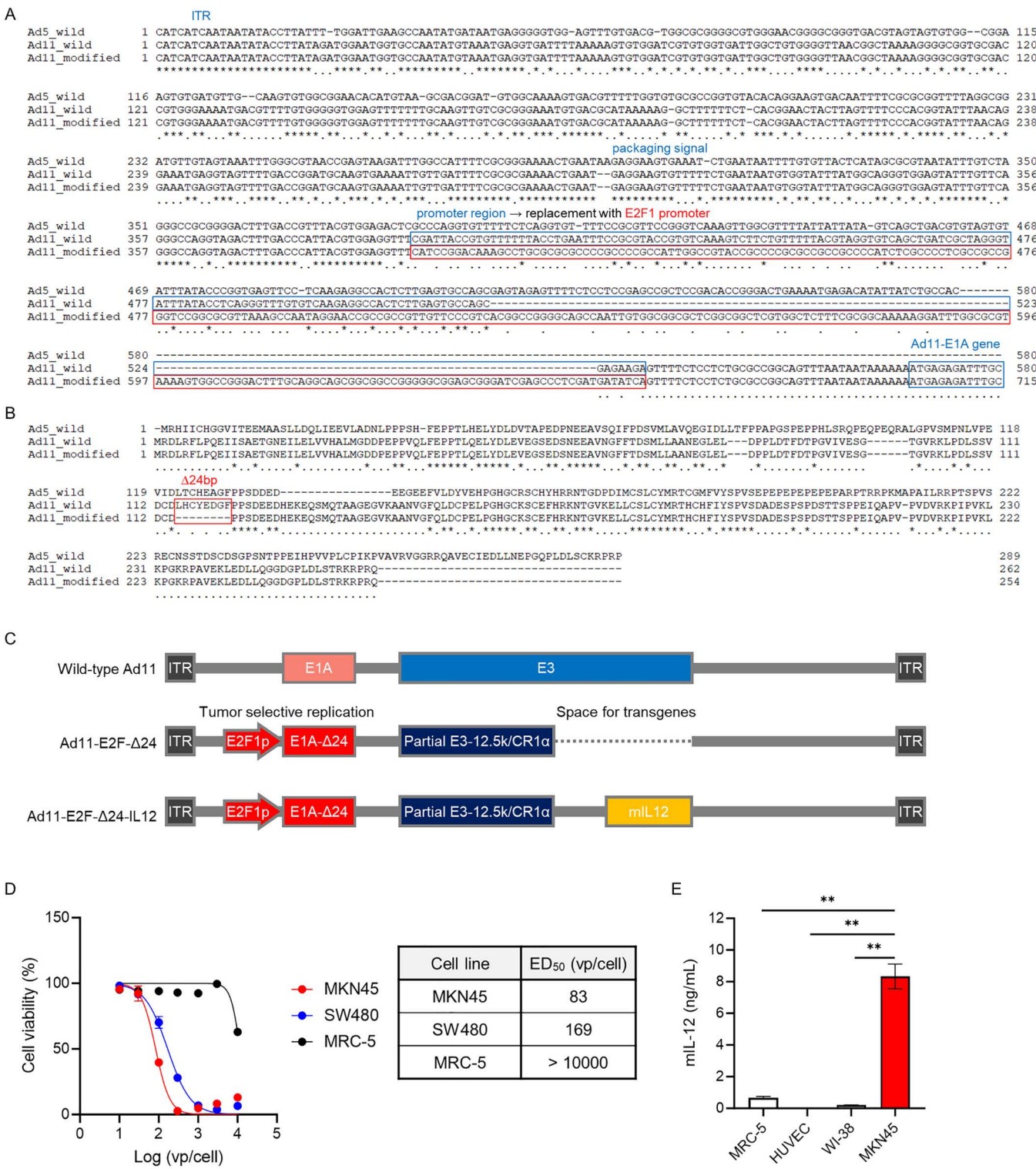

**Fig 3. Genetic modifications of Ad11 for tumor-selectivity and evaluation of tumor-selective replication. (A)** Genome sequence alignment of wild-type Ad5, wild-type Ad11, and modified Ad11 in the promoter region of the E1A gene. The endogenous promoter region of wild-type Ad11 was replaced with E2F1 promoter in the modified Ad11, resulting in its tumor-selective replication. **(B)** Amino acid sequence alignment of wild-type Ad5, wild-type Ad11, and modified Ad11 in E1A protein. An eight amino acid deletion in the modified Ad11 responsible its tumor selectivity is highlighted. **(C)**

Schematic representation of the genomic structures of wild-type Ad11, Ad11-E2F-Δ24, and Ad11-E2F-Δ24-IL12. Ad11-E2F-Δ24 was engineered for tumor-selective replication with the E2F1 promoter replacement and deletion in E1A, as described above. In addition, Ad11-E2F-Δ24 had partial deletion in the E3 region. Ad11-E2F-Δ24-IL12 included an insertion of the murine IL-12 gene into the modified E3 region. **(D)** MKN45 (human gastric cancer cells), SW480 (human colon cancer cells), and MRC5 (human normal fibroblasts) were infected with Ad11-E2F-Δ24 at different titers (vp/cell). Five days after viral infection, cytotoxicity against each cell line was measured. Table indicates the $ED_{50}$ values for each cell line. **(E)** MRC5, HUVEC (human umbilical vein endothelial cells), WI-38 (human normal fibroblasts), and MKN45 were infected with Ad11-E2F-Δ24-IL12 at 100 vp/cell in vitro. One day after viral infection, amount of IL-12 in the supernatants was quantified by ELISA. Each experiment was conducted in triplicate. Mean±SD is shown. **P < 0.01 compared with MKN45 by Dunnett's multiple comparison test.

intravenously administered Ads may be easily distributed to CD46-expressing normal cells during systemic circulation before reaching targeted tumors [31]. This assumption can be assessed using human CD46 transgenic (hCD46-tg) mice, which carry the complete human CD46 gene on a C57BL/6 background and express human CD46 protein in a pattern similar to that in humans [38] (S3 Fig). Following intravenous administration, Ad11 demonstrated a trend of higher lung targeting than Ad5 in hCD46-tg mice (6.79-fold change in mean). In contrast, lung targeting of Ad11 in the normal C57BL/6 mice was about one-order lower than that of Ad5. This suggests that the lung targeting by Ad11 in hCD46-tg mice is due to natural binding between the adenovirus and human CD46 protein expressed on the lung tissues (Figs 4B and 4C). Likewise, Ad28 showed higher lung targeting in hCD46-tg mice than in normal mice, likely due to its natural binding to CD46 protein. As expected, the amounts of Ad11 and Ad28 in the liver and spleen were higher in hCD46-tg mice than in C57BL/6 mice. In contrast, Ad5 showed much higher accumulation in the liver of both mice, likely due to its interaction with FX [28]. Intravenous administration of Ad5 resulted in acute body weight loss, elevation of plasma AST and ALT levels, and death (Fig 4D and 4E). Administration of wild-type Ad11, Ad11-E2F-Δ24 or Ad11-E2F-Δ24-IL12 induced a transient loss in body weight, which recovered within a few days. They did not cause any noticeable increase in ALT or AST.

To decrease unwanted tissue targeting of Ad11 via CD46, we thought to mask Ad11 fibers during circulation with the use of antibodies against Ad11. Through an initial screening, several monoclonal antibody clones were identified that bind to Ad11 particles and inhibit infection of Ad11 to HEK293 cells which endogenously express human CD46 (Figs 4F-4I). Among these, clones 2B5, 4G1 and 5A8 successfully suppressed binding between Ad11 and monkey CD46 (86% homologous to human CD46) expressed on monkey RBCs (Fig 4J). In this experiment, a recombinant Ad11 with a modification on the fibers to prevent binding to CD46 (Ad11-muFiber) (S4 Fig) [39,40] did not bind to monkey RBCs. 4G1 inhibited the interaction between Ad11 and human blood cells (Fig 4K), suggesting that 4G1 inhibits Ad11 binding to human CD46 as well as monkey CD46. However, clones 1A11, 3F8 and 5F6 did not inhibit binding to CD46, despite their stronger affinity for Ad11 particles (Figs 4I and 4J), suggesting that these antibodies may inhibit viral infection through a mechanism other than blocking the binding of CD46 to the virus.

Using the 4G1 clone, we examined whether the antibodies altered the biodistribution of Ad11 in hCD46-tg mice. Adenoviruses mixed with 4G1 antibodies in vitro were intravenously administered to hCD46-tg mice. The organs were then collected 3 hours later for measurement of virus copies (Fig 5A). Mixture with 4G1 decreased lung targeting of Ad11 to the level of Ad11-muFiber (Fig 5B). Mixture of Ad11 and 4G1 showed a similar antitumor efficacy to Ad11 against MKN45 in SCID mice (Figs 5C and 5D), suggesting that the inhibition of CD46 and adenoviruses binding by 4G1 appears to be transient and might not suppress infectivity to tumors.

## Discussion

This decade has seen great progress in the field of cancer immunology. Several immunotherapies have now been approved and provide clinical benefit for patients with advanced tumors [41–43]. Among these, oncolytic virotherapy is a promising cancer immunotherapy as it not only contributes to the direct killing of cancer cells but also stimulates antitumor immune responses [12]. To date, however, only T-VEC has been approved in the US, a situation which is particularly attributable to the practical difficulty of intratumoral administration and insufficient abscopal antitumor effect [10].

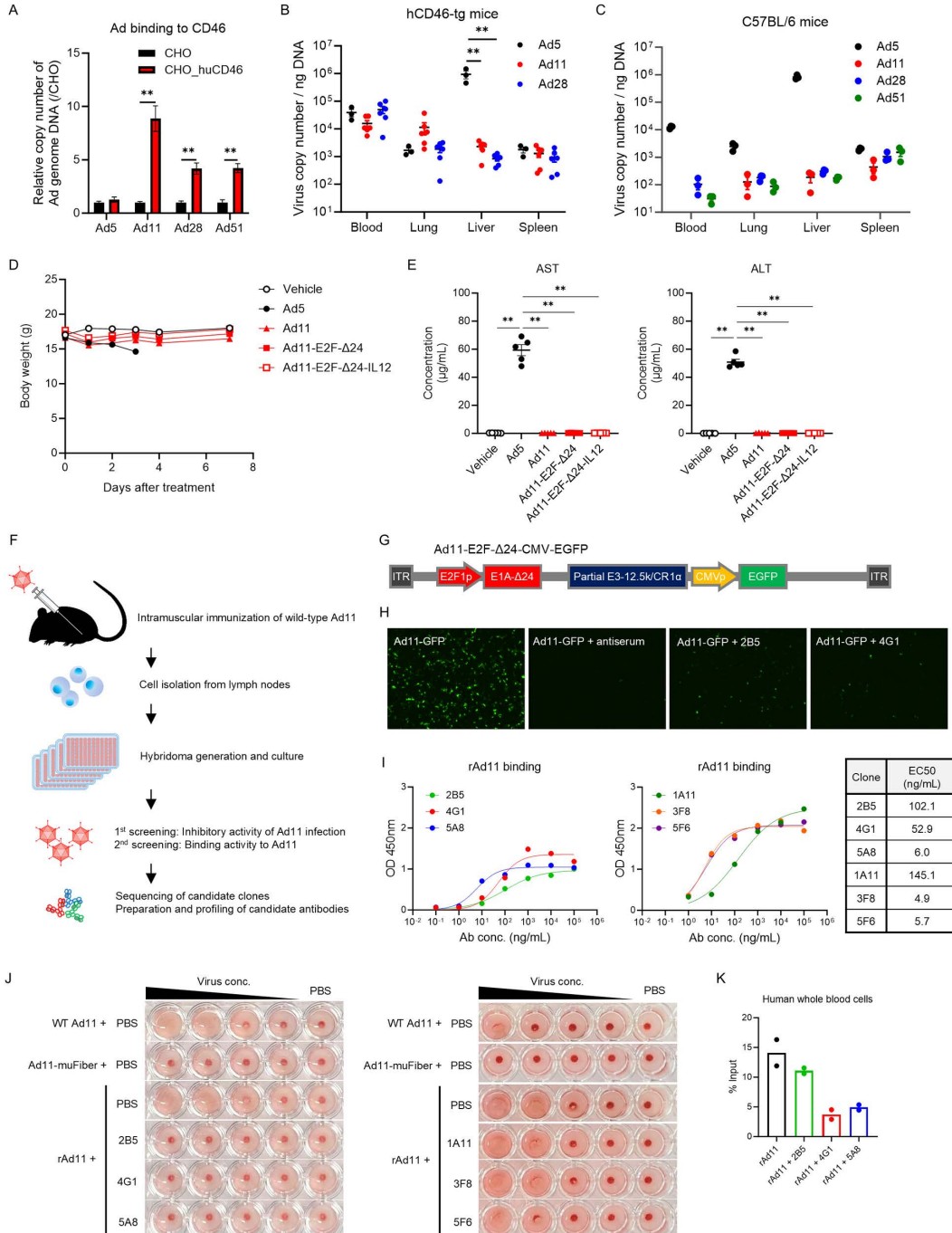

**Fig 4. Generation of anti-Ad11 monoclonal antibodies to interrupt binding to CD46. (A)** CHO cells with stable human CD46 expression were infected with wild-type adenovirus serotypes at 15 vp/cell and incubated at 4 °C for 2 hours. After incubation, the titer of adenovirus which bound to cells was evaluated by qPCR. The experiment was conducted in triplicate. Mean ± SD is shown. **$P < 0.01$ by unpaired t test. **(B and C)** CD46-tg mice ($n = 3$ for Ad5, $n = 6$ for Ad11 and Ad28) or C57BL/6 mice ($n = 3$ per group) were intravenously injected with wild-type adenovirus ($1 × 10^{10}$ vp). Forty-eight hours after injection, organs were collected. Viral DNA in each organ was measured by qPCR. Mean ± SEM is shown. **(D and E)** C57BL/6 mice were intravenously injected with viruses as indicated ($1 × 10^{10}$ vp, $n = 5$ per group). Body weight changes were monitored. Plasma AST and ALT levels 2 days post-injection were measured by ELISA. Following Ad5 administration, four mice were found moribund and euthanized upon reaching humane endpoints on day 3, and one mouse died spontaneously the following day. Mean ± SEM is shown. **$P < 0.01$ by Tukey's multiple comparison test. **(F)** Schematic representation of antibody screening. Genetically modified mice in which mouse immunoglobulins were replaced with human immunoglobulins

were intramuscularly immunized with wild-type Ad11. After immunization, mononuclear cells were isolated from lymph nodes. The cells were fused with myeloma cells to generate hybridoma cells. The culture media from hybridoma cells were used for screening assays. **(G)** Ad11-E2F-Δ24-CMV-EGFP was prepared for antibody screening. Ad11-E2F-Δ24 was inserted with an CMV-EGFP cassette into the modified E3 region. **(H)** Representative screening results of anti-Ad11 antibodies which inhibited Ad11-E2F-Δ24-CMV-EGFP (Ad11-GFP) infection of HEK293 cells. **(I)** Binding activity of purified anti-Ad11 antibodies to rAd11 was evaluated by ELISA. The experiment was conducted in duplicate. Mean is shown. Table indicates the half-maximal effective concentration (EC$_{50}$) values for each clone. **(J)** Inhibition of binding between Ad11 and monkey CD46 was assessed by hemagglutination assay. Monkey RBCs were incubated with Ad11 (wild-type Ad11, Ad11-mufiber, and rAd11) and anti-Ad11 antibody. After sedimentation of RBCs finished, hemagglutination patterns was observed. **(K)** Inhibition of binding between Ad11 and human CD46 was evaluated by whole blood binding assay. Human whole blood cells were incubated with Ad11 (wild-type Ad11, Ad11-mufiber, and rAd11) and anti-Ad11 antibody. After incubation, the titer of adenovirus which bound to cells was evaluated by qPCR. The experiment was conducted in duplicate. Mean is shown. Ad11-GFP; Ad11-E2F-Δ24-CMV-EGFP, WT Ad11; wild-type Ad11, rAd11; recombinant Ad11.

To overcome these issues, we tried to develop a recombinant oncolytic virus for intravenous administration to provide delivery to systemic metastases. The first step was to select backbone candidates which have minimum affinity to normal cells. Although our assessment identified some Ads with minimal binding to human RBCs, Ad11 showed stronger cytotoxicity against human cancer cell lines than Ad28 and Ad51, which may be attributed to the rarity and lower pathogenicity of Ad28 and Ad51 compared to Ad11 [18,19]. Following the selection of Ad11 as a backbone for further modification, the next step was to make this virus tumor-selective to minimize viral replication in normal cells. While the modification of E1A gene and integration of E2F1 promoter have been reported utilizing Ad5 [35,37], no study to date has applied these modifications to Ads in Group B, including Ad11. Our study successfully demonstrated tumor selectivity with Ad11, without compromising its cytotoxicity against cancer cells, suggesting the possibility that this system can be applied to all other Ads.

Nevertheless, the natural binding to CD46 on normal cells is a concern. Ad11 binds to CD46 via the fiber knobs and the amino acids essential to the binding have been well studied [39,40]. As expected, Ad11-muFiber, which lacks the binding sites, demonstrated reduced lung targeting after intravenous administration. However, this modification may not be suitable for oncolytic virotherapy because of its possible loss of infectivity to cancer cells. Our approach to mask the fiber knobs with specific antibodies successfully reduced lung targeting in hCD46-tg mice, likely by transiently inhibiting binding between CD46 protein on the lung and the fiber knobs, though the mechanism needs to be clarified. In oncolytic virotherapy, antibodies must be separated from Ad11 once the viruses reach the target tumor in order for the viruses to infect the tumor cells. On the other hand, sustained masking of Ad11 may be beneficial in treating respiratory diseases caused by Ad11 infection [44,45]. Although the concept of utilizing monoclonal antibodies against viral diseases is not novel [46], this study is the first to demonstrate the activity of neutralizing monoclonal antibodies in altering biodistribution in animals.

We acknowledge there are some limitations to this study that prevented full characterization of our therapeutic strategy. Our study showed that intravenous administration of Ad11 with or without 4G1 antibodies showed similar antitumor efficacy in SCID mice, suggesting that 4G1 may no longer bind to Ad11 once virus particles reach the tumors. Unfortunately, the benefit of 4G1 antibodies in antitumor efficacy against human cancer cells could not be demonstrated in hCD46-tg mice as hCD46-tg mice are immunocompetent and cannot accept human cancer cells. Furthermore, the poor susceptibility of murine cells to Ad11 prevented us from evaluating antitumor efficacy against murine cancer cell lines [47–49]. Likewise, we were unable to validate the evaluation of the 4G1 antibodies against the adenovirus infection in hCD46-tg mice. Given that our study demonstrated the function of 4G1 in inhibiting the binding between Ad11 and monkey erythrocytes, nonhuman primates are likely to be candidates for evaluation.

In conclusion, we have successfully engineered tumor-selective oncolytic Ad11 for intravenous administration and generated Ad11-specific neutralizing monoclonal antibodies to reduce unwanted lung targeting of the adenoviruses. This therapeutic strategy clearly improves the potency and safety profile of oncolytic virotherapy, albeit that further exploration is required before clinical use in patients.

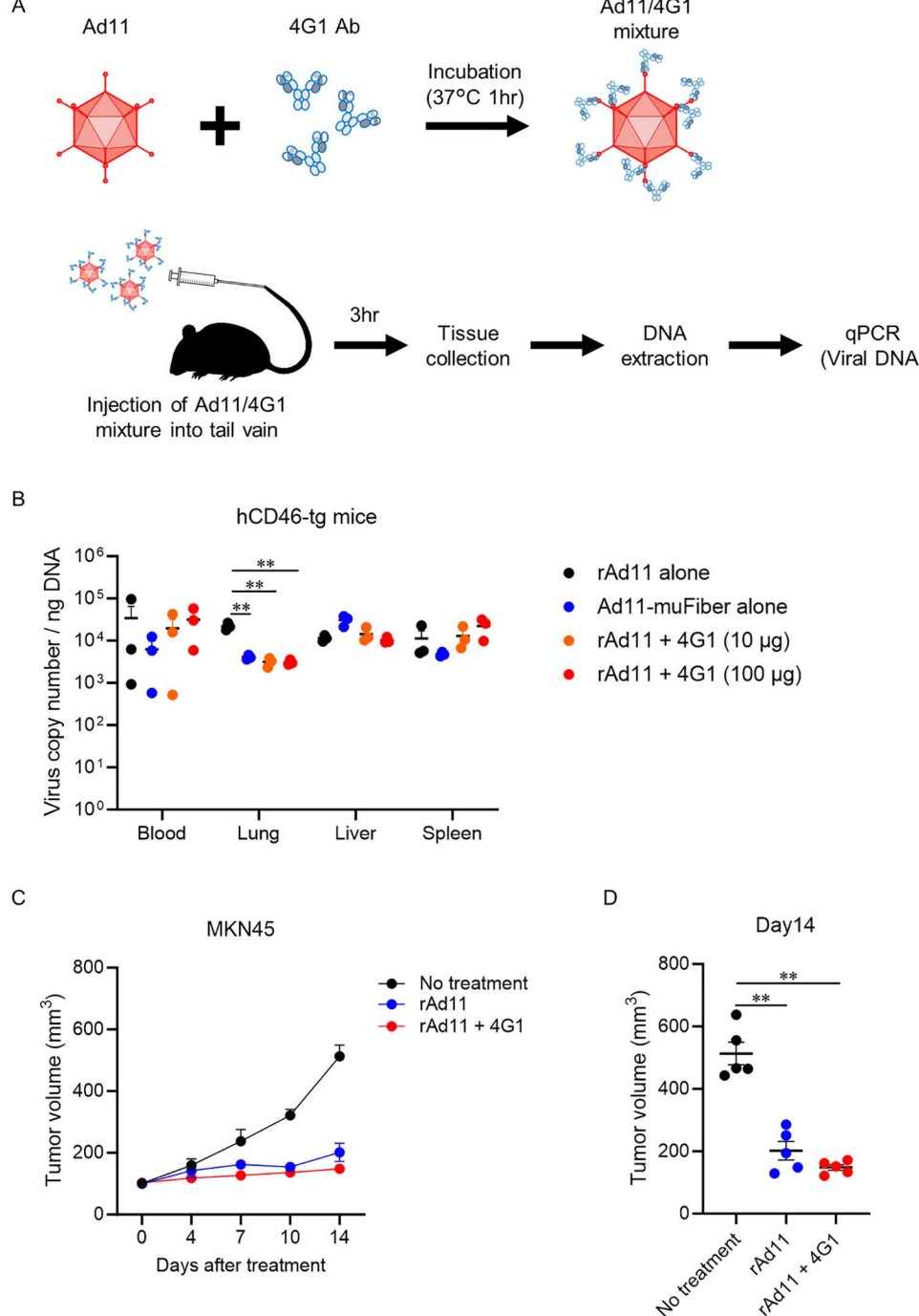

**Fig 5. Improved biodistribution of Ad11 by mixture with neutralizing antibodies. (A)** Experimental scheme. Recombinant Ad11 ($1 \times 10^{10}$ vp) was mixed with anti-Ad11 mAb 4G1 (10 or 100 µg) and then incubated at 37 °C for 1 hour. Human CD46-tg mice were intravenously injected with the mixture. Three hours after injection, organs were collected. Viral DNA in each organ was measured by qPCR. **(B)** Adenovirus biodistribution in hCD46tg mice after intravenous injection of rAd11, Ad11-muFiber, and rAd11/4G1 Ab mixture ($n=3$ per group). **$P<0.01$ compared with rAd11 alone by Dunnett's multiple comparison test. Mean±SEM is shown. **(C and D)** Mice were subcutaneously inoculated with MKN45 cells. When tumors reached about 50 to 100 mm³, tumors were intravenously injected with rAd11 ($1 \times 10^9$ vp) or rAd11/4G1 Ab mixture ($1 \times 10^9$ vp and 10 µg Ab) every other day, for a total of three injections. Tumor growth in the treated mice and tumor volume at 14 days after treatment are shown ($n=5$ per group). A total of 15 mice were used in the experiment, and no mice were euthanized or died during the 14-day study period. **$P<0.01$ by Tukey's multiple comparison test. Mean±SEM is shown.

## Supporting information

**S1 Fig.  Replication efficiency of human adenovirus serotypes in A549, Hep3B, PC-3 and HT-29 cells.** (A) Cells were infected with Ad11, Ad28, and Ad51 at 100 and 1000 vp/cell in vitro. Twenty-four and seventy-two hours after viral infection, viral genomic DNA was quantified by qPCR. The experiment was conducted in triplicate. Mean ± SD is shown. $*P < 0.05$ and $**P < 0.01$ compared with Ad11 by Dunnett's multiple comparison test. (B) Time-dependent change in virus copy number between 24 and 72 hours.
(TIF)

**S2 Fig.  Cytotoxicity of Ad11 and Ad11-E2F-Δ24 against cancer cells.** MKN45 was infected with Ad11 and Ad11-E2F-Δ24 at different titers (vp/cell). Five days after viral infection, cytotoxicity was measured. The experiment was conducted in triplicate. Mean ± SD is shown. Table indicates the $ED_{50}$ values for each cell line.
(TIF)

**S3 Fig.  Distribution of human CD46 protein in hCD46-tg mice.** Homogenates of organs from hCD46-tg mice were subjected to western blotting analysis using the indicated antibodies. Homogenates of organs from Balb/c mice were used as a negative control. Human CD46-tg mice; Tg, Balb/c mice; WT.
(TIF)

**S4 Fig.  Genomic structures of Ad11-muFiber.** For de-targeting to CD46, the CD46-binding motif in the fiber knob of Ad11-E2F-Δ24 was replaced with a GS linker.
(TIF)

**S1 Table.  Primer list used in qPCR analysis.**
(XLSX)

## Acknowledgments

We thank T. Yamazaki, Y. Oosumi, and H. Tominaga for scientific discussion, antibody screening, and antibody preparation. We thank M. Urio, T. Ohyama, and Y. Komori for technical assistance with the experiments. We thank S. Iwasaki, T. Yoshida, and R. Moriya for scientific discussion.

## Author contributions

**Conceptualization:** Masahisa Hemmi, Shinsuke Nakao.

**Data curation:** Masahisa Hemmi, Midori Yamashita, Yoshiko Shimizu, Shinsuke Nakao.

**Formal analysis:** Masahisa Hemmi, Midori Yamashita, Yoshiko Shimizu, Shinsuke Nakao.

**Investigation:** Masahisa Hemmi, Midori Yamashita, Yoshiko Shimizu.

**Methodology:** Masahisa Hemmi.

**Supervision:** Shinsuke Nakao.

**Visualization:** Masahisa Hemmi.

**Writing – original draft:** Masahisa Hemmi, Shinsuke Nakao.

**Writing – review & editing:** Masahisa Hemmi, Shinsuke Nakao.

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
