## [Decision Letter · Decision Letter 0]

Dear Dr. Nakao,

We look forward to receiving your revised manuscript.

Kind regards,

Tomoko Fujiyuki

Academic Editor

PLOS ONE

**Journal Requirements:**

2. To comply with PLOS ONE submissions requirements, in your Methods section, please provide additional information regarding the experiments involving animals and ensure you have included details on (a) methods of sacrifice, (b) methods of anesthesia and/or analgesia, and (c) efforts to alleviate suffering.

This research was funded by Astellas Pharma, Inc..

4. Thank you for stating the following in the Competing Interests: 

I have read the journal's policy and the authors of this manuscript have the following competing interests: the authors are employees of Astellas Pharma Inc., Japan.

We note that one or more of the authors are employed by a commercial company: Astellas Pharma Inc., Japan. 

“The funder provided support in the form of salaries for authors, but did not have any additional role in the study design, data collection and analysis, decision to publish, or preparation of the manuscript. The specific roles of these authors are articulated in the ‘author contributions’ section.”

Reviewers' comments:

Reviewer's Responses to Questions

**Comments to the Author**

1. Is the manuscript technically sound, and do the data support the conclusions?

Reviewer #1: Partly

Reviewer #2: Yes

2. Has the statistical analysis been performed appropriately and rigorously?

Reviewer #1: Yes

Reviewer #2: No

3. Have the authors made all data underlying the findings in their manuscript fully available?

Reviewer #1: No

Reviewer #2: Yes

4. Is the manuscript presented in an intelligible fashion and written in standard English?

Reviewer #1: Yes

Reviewer #2: Yes

**Reviewer #1: ** This is an interesting paper by Nakao et al., which seeks to explore the effect of neutralizing monoclonal antibodies on the biodistribution of intravenously administered oncolytic adenovirus in human CD46-transgenic mice.

**Reviewer #2: ** General comments:

In this manuscript, the authors evaluated several different serotypes of adenovirus (Ad) for systemic delivery application. Their findings have led to identification of serotype 11 adenovirus (Ad11) as promising candidate for systemic therapy application, as it can attenuate nonspecific hepatic sequestration associated with serotype 5 Ad (Ad5). Despite attenuation in liver tropism, systemically administered Ad11 was shown to nonspecifically accumulate at a high level in lung tissues of mice and the authors demonstrated that Ad11-specific neutralizing antibody that abrogate Ad11 binding with its cellular receptor CD46 can be used in combination with systemically administered Ad11 to attenuate nonspecific uptake in lung tissues. Although the identification of Ad serotype better suited for systemic delivery application can have important clinical relevance, there are several shortcomings that must be addressed prior to publication of the manuscript.

Specific comments:

1. In Fig. 1B, the authors fail to provide the starting viral dose and only provides dilution factor. Authors must provide exact initial viral dose and demonstrate that same quantity of viral particles were utilized across different experimental groups for the data to have significance.

2. As the authors proposed that Ad11 is a better serotype than Ad5 for oncolytic virotherapy through their findings in this study, Fig. 2 should include Ad5 as comparison group to better demonstrate how the basal cytolytic activity of Ad5 and Ad11 differ prior to genetic modification.

3. There are many different Ads utilized in this study, thus it is essential that authors provide infectious titers of the Ads utilized in the study. For example, if viral particle-to-plaque forming unit (PFU) ratio is vastly different among Ads utilized in the study, the harmonization of viral particle dose for comparative analysis can lead to incorrect data interpretation and scientific conclusion.

4. Currently, it cannot be clearly elucidated why Ad11 shows superior cytolytic effect across different cancer cell lines versus Ad26 or Ad51 based on the results from Fig. 2. It would be beneficial if authors could compare difference in replication capacity of the three Ad serotypes and provide some insight into why Ad11 exerted superior cytolytic effect over other serotype B Ads examined in this study.

5. Fig 3 would benefit from an assay comparing the cell killing effect and viral replication capacity of all three Ad11 viruses listed in Fig. 3C. These additional assays are essential to aid the readers evaluate the following points:

(1) How E2F1 promoter-based transcriptional regulation of and Rb binding site deletion to E1A gene enhanced cancer specificity of the Ad11-based oncolytic Ads used in this study

(2) How different genetic modification of viral genome (e.g., changes to E1A regulation or expression of therapeutic gene mIL12) affected key viral functions (e.g., replication capacity)

6. In general, many Figures lack any statistical comparison or analysis (e.g., Fig.2, Fig. 3, Fig 4B-C, Fig. 5C). Please perform statistical analysis and include them in the Figures to better support authors’ scientific claims in Results section.

7. In Fig 4I, there is huge discrepancies among the two duplicate sets of experiment. For example, the antibody clones 1A11, 3F8, and 5F6 do not inhibit hemolytic effect of rAd11 at the two highest viral dose levels on the experimental set provided on the right, whereas complete protection is achieved by the same antibodies on the experimental set presented on the left.

8. In line 358, the author claims that ‘clones 1A11, 3F8 and 5F6 did not inhibit binding to CD46, despite their stronger affinity for Ad11 particles (Fig 4H and 4I)’. However, there is no data that corroborate this claim, as none of the provided Figures evaluated whether 1A11, 3F8, and 5F6 inhibited CD46-mediated cell uptake of Ad11 compared with other antibody clones (2B5, 4G1, and 5A8). Please provide corresponding data to corroborate this claim.

9. There are several critical unanswered scientific questions in this study that are essential to support authors' claim that Ad11-based oncolytic therapy is better suited for systemic application than conventional Ad5.

(1) there is no direct comparison or evaluation of how rAd11 and WT Ad11 differ in biodistribution and safety profile after systemic administration in hCD46 tg mice. Systemically administered Ad11 shows either comparable or higher level of virus uptake in blood or lung tissues compared to Ad5, respectively, and these nonspecific uptake can be a safety concern.

(2) None of the biodistribution studies in hCD46 tg mice were conducted using a tumor-bearing mice. A hCD46-expressing tumor model in hCD46 tg mice should be utilized to evaluate how much of systemically administered rAd11 reaches the tumor tissues compared to normal organs in either absence or presence of 4G1 antibodies.

(3) the antitumor efficacy should also be evaluated using hCD46 tg mice model, as the current study only examined the therapeutic effect in SCID mice that does not express human CD46. As only the tumor would express human CD46, it is highly likely that nonspecific sequestration of rAd11 into normal organs and blood cells would be greatly limited, thus this model cannot be considered as an adequate model to evaluate the antitumor efficacy of rAd11. The therapeutic gene of choice for rAd11 used in antitumor efficacy study was also murine IL-12, which also requires competent immune system for accurate evaluation of safety and efficacy rather than SCID. Due to these reasons, an additional study should be performed in immune-competent hCD46 tg mice rather than SCID.

**Do you want your identity to be public for this peer review?** For information about this choice, including consent withdrawal, please see our Privacy Policy

Reviewer #1: No

Reviewer #2: No

---

## [Author Response · Author response to Decision Letter 1]

19 Feb 2025

Response to Academic Editor:

Response:

We would like to express our great appreciation for your suggestions to improve the quality of our manuscript. We have taken all suggestions into account in preparing the revised version of our manuscript and performed additional analyses. Additions to the text are shown in red and deletions are indicated with a strikethrough.

Are a* unique.

Response:

Corrected.

With the animal model in mind, it’s not clear which metastases would be implicated, lung metastases, brain?

Response:

The phrase “particularly for patients with metastases” was deleted because intravenous OV treatment is not limited to patients with metastases.

Not sure one can say “the virus was infected.”?

Response:

The sentence has been corrected. “Cells were infected with Ad11-E2F-d24-mIL12 at …”

It’s not clear in the narrative if it was DNA/RNA/etc.. that was performed using qPCR

Response:

The sentence has been revised. “… collected for qPCR analysis to detect viral DNA.”

It might be specified later, but it would be convenient to know which organs were collected.

Response:

The sentence has been revised. “… and then lung, liver, spleen and blood were collected.”

What does conventional methods mean? I think that this section could be more detailed, the specificity of the monoclonal antibodies is at the core of the paper’s hypothesis of reducing toxicity, however, the authors provide little explanation as to how the antibodies were selected, and how inhibition of Ad11 was identified.

Response:

According to the suggestion, we added sentences in this section (Materials & Methods) to explain the generation and screening of hybridomas.

Which organs? Specify

Response:

The sentence has been revised. “Lung, liver and spleen collected from mice were…”

Which proteins were measured here? It’s not clear.

Response:

The sentence has been revised. “The concentrations of total protein in these samples were…”

Given the importance of the selection of which adenovirus would have the least affinity, there seems to be very little detail provided about the pre-exposure of the donors to adenovirus, with a number (n=5) that might not be representative of total population affinity characteristics.

Response:

According to the comment, the sentence has been changed in the Result section. The purpose of this experiment was to prioritize serotypes and exclude those that may bind to human RBCs, considering that the binding to RBCs would reduce the delivery efficiency after intravenous administration. We don’t intend to note that RBCs from all humans bind Ad5 or that RBCs from all humans do not bind Ad11. Additionally, the following comment was added in the Materials & Methods section; “No information was available on antibody titers for each serotype”.

In this context, it could be highlighted which receptor is used by Ad5.

Response:

We added “Coxsackievirus-Adenovirus Receptor (CAR)” to explain the receptor of Ad5.

A dot seems to be missing here.

Response:

Thank you. Corrected.

In light of the results stipulated in the manuscript regarding the cytotoxicity Ad11, I would think it’s important to show statistical assessments here. Also, the number of days should be indicated on the graphs. The absence of a negative is also noted.

Response:

ED50 values of each Ad for each cell line were added on Figure 2. The number of days (4 days post infection) was indicated on each graph. Cell viability was calculated by setting uninfected cells (negative control) and medium control wells (positive control) to 100% and 0% survival, respectively. This sentence was added in the Materials & Methods section.

Within the same reasoning and considering the absence of any statistical tests that would reinforce this hypothesis, the strength of cytotoxicity cannot be evaluated.

Response:

The highlighted phrase was deleted.

No statistical tests have been performed to test the hypothesis regarding cytotoxicity in fibroblasts vs colorectal nor adenocarcinoma cell lines.

Response:

ED50 values for each cell were added on Figure 3. Statistical analysis was conducted on Figure 3E to show cancer-selective virus activity.

This statement has to be supported by statistical tests and there are no details about how this secretion was tested, through infection in vitro? There is a lack of details regarding the process of the experiment that led to this conclusion.

Response:

Detailed description including the statistical analysis was added in the figure legend.

Absence of statistical testing to support the claims. Typo: Relative*. Axis title could be renamed to reflect what was measured, and the nature of the qPCR performed. The sample size seems small and would be limiting the statistical power. (n=3?) Also, it is not clear from the narrative why a smaller number of mice were used for Ad5?

Response:

According to the suggestion, statistical analysis was performed on Figure 4A. The figure was revised. Regarding the data on Figure 4B, texts in the Result section were changed. The limited number of mice in the Ad5 group was due to a shortage in supply of CD46-tg mice.

These results depend mainly on the tested results on HEK293, while there is no mention of the choice for this cell line, instead of lung or ovarian cancer cell line that were mentioned previously.

Response:

The HEK293 cell line is highly sensitive to Ad11, making it suitable for the propagation and purification of Ad11 (and recombinant viruses). Due to this sensitivity, HEK293 was chosen for the hybridoma screening.

In this graph, the is absence of non-specific control is noted, which would be needed to corroborate the effectiveness of 4G1. Were the results statistically analysed to determine significance? Is there any suggestion of dose-response to 4G1 that determines the dose-response relationship and antibody’s potency?

Response:

Figure 4G was deleted. Dose dependency of 4G1 antibody as well as other antibodies is described in former Figure 4H.

Statistical significance?

Response:

Since this experiment was not intended for prioritization of antibody clones, statistical analysis was not conducted. Instead, EC50 values were added.

Incubation at 37°C and only an hour is not justified here, ideally, there would be a time-course study to indicate the optimal incubation time. Unless previously performed.

Response:

We appreciate your insightful comment. B. Dreier et al., incubated a recombinant Ad5 with its adapter protein for 1 hour at room temperature (PNAS, 2013). Whole blood binding assay in this study (Figure 4I, 4G1 binding to rAd11) was performed with 1 hour incubation at 37°C. Likewise, we chose “1 hour incubation at 37°C” before animal studies.

The authors show only mice that were treated, there are no control groups or non-specific antibodies.

Response:

This comment is also appreciated. This experiment was performed after the preliminary experiments to observe dose-dependency of Ad11. We did not include non-specific antibodies because this experiment is to demonstrate “non-effectiveness” of 4G1 antibody (4G1 does not interfere Ad11 infection in tumor). We acknowledge further experiments with use of more suitable animal models are essential to demonstrate the benefit of this antibody, as described in the Discussion section.

This transient nature of the antibody binding needs to be investigated further in matters of reversibility and duration of effect.

Response:

We added a sentence in the Discussion section to note the necessity of the series of additional mechanism analysis. Various hypotheses can be considered, the nature of the antibody, effect of tumor microenvironment, etc.

Response to Reviewer #1:

Reviewer #1: Comments to the Author

This is an interesting paper by Nakao et al., which seeks to explore the effect of neutralizing monoclonal antibodies on the biodistribution of intravenously administered oncolytic adenovirus in human CD46-transgenic mice.

Response:

We are pleased to hear your supportive comment. The manuscript was revised according to the Editor’s and another Reviewer’s comments. Additions to the text are shown in red and deletions are indicated with a strikethrough.

 

Response to Reviewer #2:

Reviewer #2: General comments:

In this manuscript, the authors evaluated several different serotypes of adenovirus (Ad) for systemic delivery application. Their findings have led to identification of serotype 11 adenovirus (Ad11) as promising candidate for systemic therapy application, as it can attenuate nonspecific hepatic sequestration associated with serotype 5 Ad (Ad5). Despite attenuation in liver tropism, systemically administered Ad11 was shown to nonspecifically accumulate at a high level in lung tissues of mice and the authors demonstrated that Ad11-specific neutralizing antibody that abrogate Ad11 binding with its cellular receptor CD46 can be used in combination with systemically administered Ad11 to attenuate nonspecific uptake in lung tissues. Although the identification of Ad serotype better suited for systemic delivery application can have important clinical relevance, there are several shortcomings that must be addressed prior to publication of the manuscript.

Response:

We wish to express our great appreciation for your insightful suggestions to improve our manuscript. As indicated in the responses that follow, we have taken all suggestions into account in preparing the revised version of our manuscript. We performed additional experiments and additional analyses. Additions to the text are shown in red and deletions are indicated with a strikethrough.

Specific comments:

1. In Fig. 1B, the authors fail to provide the starting viral dose and only provides dilution factor. Authors must provide exact initial viral dose and demonstrate that same quantity of viral particles were utilized across different experimental groups for the data to have significance.

Response:

In this experiment same quantity of viral particles were utilized across the different serotypes. The starting viral dose (1x109 vp) was added in the figure.

2. As the authors proposed that Ad11 is a better serotype than Ad5 for oncolytic virotherapy through their findings in this study, Fig. 2 should include Ad5 as comparison group to better demonstrate how the basal cytolytic activity of Ad5 and Ad11 differ prior to genetic modification.

Response:

Given the binding property of Ad5 to human RBCs (Fig. 1B), we excluded Ad5 from our candidates, regardless of its potent cytotoxicity against cancer cells. E.V. Shashkova et al. reported higher cytotoxicity of Ad5 than Ad11 in most cases (Virology, 2009). H.H. Wong et al. also demonstrated higher activity of Ad5, though some cancer cell lines showed higher sensitivity to Ad11 (Molecular Therapy, 2012).

3. There are many different Ads utilized in this study, thus it is essential that authors provide infectious titers of the Ads utilized in the study. For example, if viral particle-to-plaque forming unit (PFU) ratio is vastly different among Ads utilized in the study, the harmonization of viral particle dose for comparative analysis can lead to incorrect data interpretation and scientific conclusion.

Response:

The ratio of vp/TCID50 for Ad11, Ad28 and Ad51 was 1:1.3:0.3. This information, which was added in the Result section, supports our conclusion that Ad11 has higher cytotoxicity than Ad28 or Ad51 in most cases (Figure 2).

4. Currently, it cannot be clearly elucidated why Ad11 shows superior cytolytic effect across different cancer cell lines versus Ad26 or Ad51 based on the results from Fig. 2. It would be beneficial if authors could compare difference in replication capacity of the three Ad serotypes and provide some insight into why Ad11 exerted superior cytolytic effect over other serotype B Ads examined in this study.

Response:

According to the Reviewer’s suggestion, we measured virus copy numbers 24 and 72 hours post infection with Ad11, Ad28 or Ad51. Consistent with the higher cytotoxicity of Ad11, Ad11 replicated in A549 cells more efficiently than others. On the other hand, Ad11 was lower than others in Hep3B cells, suggesting that in vitro cytotoxicity of Ads did not correlate with the replication capacity (Supplementary figure 1).

5. Fig 3 would benefit from an assay comparing the cell killing effect and viral replication capacity of all three Ad11 viruses listed in Fig. 3C. These additional assays are essential to aid the readers evaluate the following points:

(1) How E2F1 promoter-based transcriptional regulation of and Rb binding site deletion to E1A gene enhanced cancer specificity of the Ad11-based oncolytic Ads used in this study

(2) How different genetic modification of viral genome (e.g., changes to E1A regulation or expression of therapeutic gene mIL12) affected key viral functions (e.g., replication capacity)

Response:

According to the Reviewer’s comments, we compared the in vitro cytotoxicity of wild-type and the recombinant Ads and found that the modifications did not reduce viral fitness (Supplementary figure 2).

6. In general, many Figures lack any statistical comparison or analysis (e.g., Fig.2, Fig. 3, Fig 4B-C, Fig. 5C). Please perform statistical analysis and include them in the Figures to better support authors’ scientific claims in Results section.

Response:

According to the Reviewer’s comments, statistical analysis results were added on Figure 3, 4 and 5. ED50 values were added on Figure 2, 3 and 4.

7. In Fig 4I, there is huge discrepancies among the two duplicate sets of experiment. For example, the antibody clones 1A11, 3F8, and 5F6 do not inhibit hemolytic effect of rAd11 at the two highest viral dose levels on the experimental set provided on the right, whereas complete protection is achieved by the same antibodies on the experimental set presented on the left.

Response:

Group B Ads are reported to use CD46 as their primary virus attachment receptor, while virus internalization occurs thorough penton-integrin interactions (D.M. Shayakhmetov, et al., Journal of Virology, 2005). Therefore, we hypothesized that the 1A11-group (including 3F8 and 5F6) binds to the pentons to inhibit virus internalization whereas the 4G1-group (including 2B5 and 5A8) binds to the fiber knobs to inhibit interaction with CD46. Though several experiments are ongoing, the hypothesis has not yet proven due to some technical issues.

8. In line 358, the author claims that ‘clones 1A11, 3F8 and 5F6 did not inhibit binding to CD46, despite their stronger affinity for Ad11 particles (Fig 4H and 4I)’. However, there is no data that corroborate this claim, as none of the provided Figures evaluated whether 1A11, 3F8, and 5F6 inhibited CD46-mediated cell uptake of Ad11 compared with other antibody clones (2B5, 4G1, and 5A8). Please provide corresponding data to corroborate this claim.

Response:

Experiments in Figure 4I are for the assessment of CD46-Ad11 binding, given that wild-type Ad11 did show hemagglutination whereas Ad11-muFiber which lacks CD46 binding sites did not.

9. There are several critical unanswered scientific questions in this study that are essential to support authors' claim that Ad11-based oncolytic therapy is better suited for systemic application than conventional Ad5.

(1) there is no direct comparison or evaluation of how rAd11 and WT Ad11 differ in biodistribution and safety profile after systemic administration in hCD46 tg mice. Systemically administered Ad11 shows either comparable or higher level of virus uptake in blood or lung tissues compared to Ad5, respectively, and these nonspecific uptake can be a safety concern.

Response:

We appreciate the insightful comment. As described in the Discussion section, we were unable to evaluate the safety of rAds in hCD46-tg mice, due to the poor susceptibility of murine cells to Ad11. Monkey is a candidate which has susceptibility to Ad11.

(2) None of the biodistribution studies in hCD46 tg mice were conducted using a tumor-bearing mice. A hCD46-expressing tumor model in hCD46 tg mice should be utilized to evaluate how much o

---

## [Decision Letter · Decision Letter 1]

Dear Dr. Nakao,

We look forward to receiving your revised manuscript.

Kind regards,

Tomoko Fujiyuki

Academic Editor

PLOS ONE

**Additional Editor Comments:**

Although it has been greatly improved, one reviewer still has two concerns, particularly regarding safety. The authors need to address these comments. 

Reviewers' comments:

Reviewer's Responses to Questions

**Comments to the Author**

Reviewer #2: (No Response)

2. Is the manuscript technically sound, and do the data support the conclusions?

Reviewer #2: Partly

3. Has the statistical analysis been performed appropriately and rigorously?

Reviewer #2: Yes

4. Have the authors made all data underlying the findings in their manuscript fully available?

Reviewer #2: (No Response)

5. Is the manuscript presented in an intelligible fashion and written in standard English?

Reviewer #2: Yes

Reviewer #2: General Comment:

The authors have addressed most of the requests and issues suggested by the Reviewer during the revision, but there are few critical issues that persist after the initial revision process.

Thus, the manuscript is not suitable for publication in the current state.

Specific Comments:

1. In response to Q4 by Reviewer #2, the authors reported that the superior cell killing effect of Ad11 over Ad28 or Ad51 in Figure 2 did not correlate with their replication capacity based on the results from newly added Supplementary Figure S1. However, this conclusion made by the authors seems premature and inadequate, since the data clearly demonstrated that time-dependent change in virus copy number between 24 and 72 hours is notably higher in Ad11 compared to other two Ad serotypes in both A549 and Hep3B cells at all dose levels.Specifically, the fold change in viral titer in both A549 and Hep3B cells from 24h to 72h after inflection with Ad11 is nearly 100-fold (2 log increase), whereas the both Ad28 and Ad51 only exhibit around 10-fold (1 log) increase in viral quantity. These results would indicate that Ad11 generated nearly 10-fold more progeny virus over 48 h period than Ad28 or Ad51, suggesting more effective replication capacity. The lower level of Ad11 virus copy number compared to Ad28 or Ad51 in Hep3B cells at 72 hours after infection at 1,000 vp/cell were likely due to these cells being less susceptible to cellular internalization of Ad11 compared to the other two adenovirus serotypes, as suggested by nearly 10-fold lower viral quantity being detected at early time after the infection (24 h). The difference in final viral quantity at 72 hours after the infection by Ad11 versus Ad28/Ad51 is markedly lower than those observed at 24 hours after infection. Due to these reasons, it is premature to conclude that superior cytolytic effect of Ad11 was not achieved through higher replication efficiency.

Thus, additional experiments evaluating viral replication capacity in few other cancer cell lines and cellular uptake efficiency of Ad11, Ad28, and Ad51 would be able to more clearly elucidate whether the superior cytotoxicity of Ad11 was either replication-dependent or -independent.

2. Authors' response to Reviewer #2's Q9 that requested for safety and efficacy evaluation of their oncolytic adenovirus candidate in hCD46tg mice model is not adequate. Authors have stated "poor susceptibility of murine cells to Ad11" as the main reason for not performing safety assessment in mouse models, but there are several issues with this response.

Firstly, the main scope of this study was to identify alternative Ad serotype that is more suitable for intravenous administration application than Ad5, as authors noted that nonspecific liver sequestration and toxicity along with hemolytic side effect of Ad5 as major limitations of Ad5 for intravenous administration. As authors have highlighted several safety concerns regarding Ad5 as an intravenous administrable therapeutic platform, even a preliminary safety assessment of their newly developed intravenously administrable oncolytic adenovirus platform should be provided.

Secondly, authors have noted that monkeys are a much better model for safety analysis, and we agree with this point. However, no safety assessment data in monkeys have been provided by the authors during the revision, and clinical relevance of mice model for preliminary safety assessment of intravenously administered adenovirus should not be completely overlooked. For example, it is extremely common for reports investigating intravenously administered Ad5-based therapy to include safety analysis data from mice before they advance to more clinically relevant animal models like Syrian hamsters or Monkeys (Commun Biol. 2024 Sep 13;7(1):1132. Mol Ther Oncolytics. 2017 Oct 26:7:76-85. J Immunother Cancer. 2021 Nov 9;9(11):e003254.), despite the mice model being poorly permissive and susceptible to Ad5 infection in similar manner to Ad11. Additionally, it is important to note that many of the toxicities observed in mice models were also recapitulated in human patients (e.g., immune-related adverse events and liver toxicity).

Lastly, authors have chosen to utilize mouse IL-12 as therapeutic gene of choice in their oncolytic adenovirus candidate and assessment of its effect would be most relevant in mice models rather than monkey or humanized models. As IL12, a potent pro-inflammatory cytokine, could exacerbate immune-related adverse events after systemic administration, it is essential to evaluate whether this transgene can be safely expressed without significant off-target toxicity after intravenous administration by oncolytic adenovirus.

In absence of presentable safety analysis data from monkey, it is essential for authors to investigate preliminary safety profile of their IL12-expressing oncolytic adenovirus after systemic administration to better support their claim that Ad11-based oncolytic virotherapy is a more suitable candidate for intravenous therapy than conventional Ad5.

**Do you want your identity to be public for this peer review?** For information about this choice, including consent withdrawal, please see our Privacy Policy

Reviewer #2: No

---

## [Author Response · Author response to Decision Letter 2]

22 May 2025

Response to Reviewer #2:

Reviewer #2: General comments:

The authors have addressed most of the requests and issues suggested by the Reviewer during the revision, but there are few critical issues that persist after the initial revision process.

Thus, the manuscript is not suitable for publication in the current state.

Specific Comments:

1. In response to Q4 by Reviewer #2, the authors reported that the superior cell killing effect of Ad11 over Ad28 or Ad51 in Figure 2 did not correlate with their replication capacity based on the results from newly added Supplementary Figure S1. However, this conclusion made by the authors seems premature and inadequate, since the data clearly demonstrated that time-dependent change in virus copy number between 24 and 72 hours is notably higher in Ad11 compared to other two Ad serotypes in both A549 and Hep3B cells at all dose levels. Specifically, the fold change in viral titer in both A549 and Hep3B cells from 24h to 72h after inflection with Ad11 is nearly 100-fold (2 log increase), whereas the both Ad28 and Ad51 only exhibit around 10-fold (1 log) increase in viral quantity. These results would indicate that Ad11 generated nearly 10-fold more progeny virus over 48 h period than Ad28 or Ad51, suggesting more effective replication capacity. The lower level of Ad11 virus copy number compared to Ad28 or Ad51 in Hep3B cells at 72 hours after infection at 1,000 vp/cell were likely due to these cells being less susceptible to cellular internalization of Ad11 compared to the other two adenovirus serotypes, as suggested by nearly 10-fold lower viral quantity being detected at early time after the infection (24 h). The difference in final viral quantity at 72 hours after the infection by Ad11 versus Ad28/Ad51 is markedly lower than those observed at 24 hours after infection. Due to these reasons, it is premature to conclude that superior cytolytic effect of Ad11 was not achieved through higher replication efficiency.

Thus, additional experiments evaluating viral replication capacity in few other cancer cell lines and cellular uptake efficiency of Ad11, Ad28, and Ad51 would be able to more clearly elucidate whether the superior cytotoxicity of Ad11 was either replication-dependent or -independent.

Response:

According to the Reviewer’s suggestion, we further evaluated virus replication in PC-3 and HT-29 cells. Our results showed that Ad11 exhibited consistently higher increase in viral copy numbers across all tested cell lines (Supplementary figure 1). These findings suggests that the enhanced virus replication may contribute to the superior cytotoxicity of Ad11, whereas other mechanisms may also be involved in its potent killing activity, as reported by Wong et al. (Molecular Therapy, 2012). The results have been incorporated, and the conclusion has been revised accordingly.

2. Authors' response to Reviewer #2's Q9 that requested for safety and efficacy evaluation of their oncolytic adenovirus candidate in hCD46tg mice model is not adequate. Authors have stated "poor susceptibility of murine cells to Ad11" as the main reason for not performing safety assessment in mouse models, but there are several issues with this response.

Firstly, the main scope of this study was to identify alternative Ad serotype that is more suitable for intravenous administration application than Ad5, as authors noted that nonspecific liver sequestration and toxicity along with hemolytic side effect of Ad5 as major limitations of Ad5 for intravenous administration. As authors have highlighted several safety concerns regarding Ad5 as an intravenous administrable therapeutic platform, even a preliminary safety assessment of their newly developed intravenously administrable oncolytic adenovirus platform should be provided.

Secondly, authors have noted that monkeys are a much better model for safety analysis, and we agree with this point. However, no safety assessment data in monkeys have been provided by the authors during the revision, and clinical relevance of mice model for preliminary safety assessment of intravenously administered adenovirus should not be completely overlooked. For example, it is extremely common for reports investigating intravenously administered Ad5-based therapy to include safety analysis data from mice before they advance to more clinically relevant animal models like Syrian hamsters or Monkeys (Commun Biol. 2024 Sep 13;7(1):1132. Mol Ther Oncolytics. 2017 Oct 26:7:76-85. J Immunother Cancer. 2021 Nov 9;9(11):e003254.), despite the mice model being poorly permissive and susceptible to Ad5 infection in similar manner to Ad11. Additionally, it is important to note that many of the toxicities observed in mice models were also recapitulated in human patients (e.g., immune-related adverse events and liver toxicity).

Lastly, authors have chosen to utilize mouse IL-12 as therapeutic gene of choice in their oncolytic adenovirus candidate and assessment of its effect would be most relevant in mice models rather than monkey or humanized models. As IL12, a potent pro-inflammatory cytokine, could exacerbate immune-related adverse events after systemic administration, it is essential to evaluate whether this transgene can be safely expressed without significant off-target toxicity after intravenous administration by oncolytic adenovirus.

In absence of presentable safety analysis data from monkey, it is essential for authors to investigate preliminary safety profile of their IL12-expressing oncolytic adenovirus after systemic administration to better support their claim that Ad11-based oncolytic virotherapy is a more suitable candidate for intravenous therapy than conventional Ad5.

Response:

We appreciate insightful suggestions from the Reviewer. Based on previous reports investigating the safety of intravenously administered Ad5, we assessed the replication of Ad11 in cells derived from mouse, hamster, and monkey. Our results showed that Ad11 did not replicate in mouse and hamster cell lines, that is the reason why we did not pursue mouse models previously, although we acknowledge their clinical relevance for safety assessment.

In this revision, we evaluated the safety of wild-type Ad5, wild-type Ad11, and recombinant Ad11s in mice. Intravenous administration of Ad5 resulted in acute body weight loss and elevation of plasma AST and ALT levels. Intravenous administration of Ad11s did not increase AST or ALT levels, while they induced a transient body weight loss, which resolved within a few days (Figure 4D and 4E).

---

## [Decision Letter · Decision Letter 2]

Neutralizing monoclonal antibodies improve biodistribution of intravenously administered oncolytic adenovirus in human CD46-transgenic mice

PONE-D-24-49759R2

Dear Dr. Nakao,

We’re pleased to inform you that your manuscript has been judged scientifically suitable for publication and will be formally accepted for publication once it meets all outstanding technical requirements.

Kind regards,

Tomoko Fujiyuki

Academic Editor

PLOS ONE

Additional Editor Comments (optional):

Reviewers' comments:

Reviewer's Responses to Questions

**Comments to the Author**

Reviewer #2: All comments have been addressed

2. Is the manuscript technically sound, and do the data support the conclusions?

Reviewer #2: Yes

3. Has the statistical analysis been performed appropriately and rigorously?

Reviewer #2: I Don't Know

4. Have the authors made all data underlying the findings in their manuscript fully available?

Reviewer #2: Yes

5. Is the manuscript presented in an intelligible fashion and written in standard English?

Reviewer #2: Yes

Reviewer #2: (No Response)

**Do you want your identity to be public for this peer review?** For information about this choice, including consent withdrawal, please see our Privacy Policy

Reviewer #2: No

---

## [Editor Report · Acceptance letter]

PONE-D-24-49759R2

PLOS ONE

Dear Dr. Nakao,

I'm pleased to inform you that your manuscript has been deemed suitable for publication in PLOS ONE. Congratulations! Your manuscript is now being handed over to our production team.

Kind regards,

on behalf of

Dr. Tomoko Fujiyuki

Academic Editor

PLOS ONE